# Analysis of Spatiotemporal Characteristics and Influencing Factors for the Aid Events of COVID-19 Based on GDELT

**Yunxing Yao, Yinbao Zhang \*, Jianzhong Liu \*, Yanpei Li [1] and Xiaopei Li**

School of Geoscience and Technology, Zhengzhou University, Zhengzhou 450001, China
\* Correspondence: zhangyinbao@zzu.edu.cn (Y.Z.); liujianzhong@zzu.edu.cn (J.L.)

**Abstract:** The uncertainty of COVID-19 and the spatial inequality of anti-pandemic materials have made international aid an important means for many countries to cope with this global public health crisis. It is of far-reaching significance to analyze the spatiotemporal characteristics and influencing factors of international aid events for the global joint fight against COVID-19 and the sustainability of global public health business. The data on aid events from 23 January 2020 to 31 October 2021, were from the GDELT database. China, the United States, the United Kingdom, and Canada were selected as the study objects because they provided more aid. Their spatiotemporal characteristics of main aid flows, the response characteristics of the aid requests, and the characteristics of verbal aid to cash in were studied using spatial statistical analysis methods. The influencing factors of aid allocation also were studied by regression analysis. The results found that: the international aid flow of each country was consistent in spatial distribution, mainly to countries with severe pandemics and neighboring countries. However, there were differences in the recipients. China mainly aided developing countries, while the United States, the United Kingdom, and Canada mainly aided developed countries. Relatively speaking, China was more responsive to aid requests and more aggressive in cashing in on verbal aid. The countries considered the impact of their economic interests when they planned to aid. At the same time, there were obvious "bandwagon effect" and "small country tendency" on the aid events.

**Keywords:** COVID-19; international aid; spatiotemporal characteristics; influencing factors; response degree

## 1. Introduction

The outbreak and spread of COVID-19 in the world have brought a great blow to the safety of people's lives and property, as well as the political and economic development of countries around the world [1]. In 2018, according to *The Lancet* report on *Healthcare Access and Quality* (HAQ) in 195 countries worldwide, the countries with high HAQ index were mainly in Europe, while most countries in Africa were ranked lower [2]. This fully reflected the unequal distribution of medical resources in countries worldwide. In such a situation, shortage of anti-pandemic materials, lack of experience, and increasing mobility of people have become problems and obstacles for each country to fight the pandemic. Controlling the outbreak and spread of the pandemic in a short time was a great challenge for any country. Therefore, international aid has become an important means for many countries to fight this pandemic. The study of international aid driven by the pandemic can provide reference for countries to make aid policies. At the same time, it will also help to promote donors to provide aid more equitably and effectively so that international aid can play a greater role in alleviating the difficulties of the pandemic.

The *Encyclopedia Britannica* has described international aid as an international transfer of goods, services, or capital from a donor country or organization to a recipient country, based on the interests of the recipient country or its people. International aid after World War II was mainly provided by developed Western countries in the form of emergency

humanitarian aid, military aid, and economic development aid, etc. In the 1970s, international aid also became popular in some rich developing countries, such as China, India, and Arab countries [3]. Many of the current studies on international aid were based on economics, social science, and politics. In addition, the sources of aid data were mostly based on the AidData database, the official development assistance database of OECD (Organization for Economic Cooperation and Development), and media reports on international aid. Overall, current studies on international aid mainly focused on foreign aid policies [4,5], aid allocation [6,7], and aid effects [8,9]. Since the global outbreak of COVID-19, international aid events driven by the pandemic have attracted the attention of scholars. Studies have focused on several aspects, such as the impact of the pandemic on countries' aid policies [10,11], suggestions for international aid driven by the pandemic [12], public attitudes toward international aid [13,14], and aid effectiveness [15]. In addition, Bugra [16] used geographical methods to study the spatial distribution of international aid driven by the pandemic in Turkey and analyzed the influencing factors. China's international aid has become the focus of many scholars due to its early control of the pandemic and its foreign aid. Pyo [17] believed that in the background of COVID-19, China's foreign aid enhanced its international reputation and promoted the development of global public health business. However, Qi [18] found that European media believed that China's aid to Europe was compensation for the first outbreak of the pandemic in China, based on the long-standing stereotype of China.

The GDELT (Global Database of Event, Language, Tone) database was used to obtain data on international events driven by COVID-19. It records news events in print, broadcast, and webform in more than 100 languages all the time, and has attracted more and more scholars' attention in recent years. The database has been frequently used to conduct studies on geo-relations [19,20], bilateral relations [21], and political-strategic risk assessment [22]. It has also been applied to the estimation of social indicators [23], the prediction of social phenomena [24], and the assessment of the impact of international events [25]. Meanwhile, the GDELT database has also been used to study the evolution of international relations during the pandemic [26]. Methods commonly used to study GDELT include complex networks [27], event data analysis [28], regression models [29], hidden Markov [30], and frequent subgraphs [31], etc.

In general, the current studies on international aid were mostly from the perspectives of economics, social science, and politics. However, there were few studies on international aid driven by the pandemic from the perspective of geography.

For this reason, this paper extracted international aid events driven by the pandemic from the GDELT event database, and selected donor countries with a high number of aid events as objects of the study. Firstly, we analyzed the spatiotemporal characteristics of aid flows and sources of aid during the pandemic; we also built a mathematical model to analyze the response to aid requests and the characteristics of verbal aid to cash by donor countries. Then a regression analysis of the influencing factors of aid allocation during the pandemic was conducted. The purpose was to explore the pattern of international aid during the pandemic and to support future aid allocation and policy making for countries.

## 2. Data and Methods

The process of data acquisition and processing, spatiotemporal characteristics and influencing factors analysis of COVID-19 aid events in this paper are shown in Figure 1.

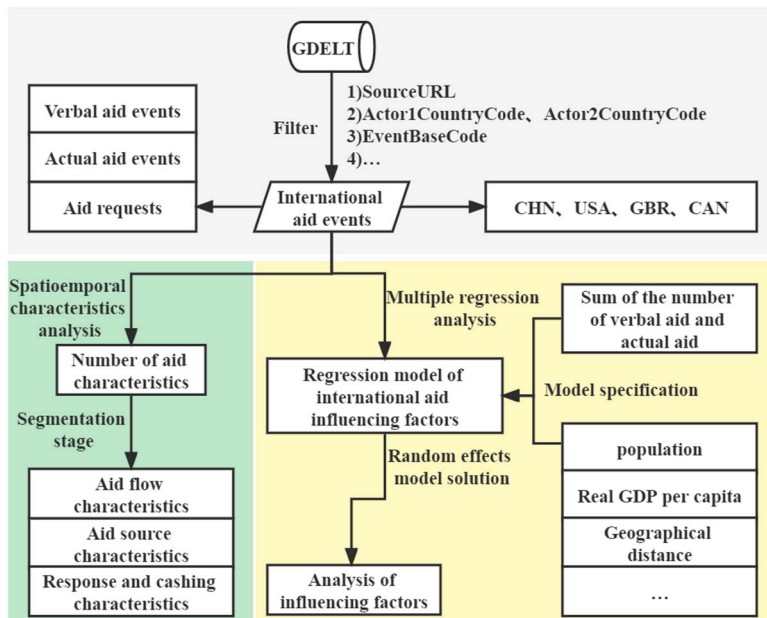

**Figure 1.** Flow chart of data acquisition, characterization and influencing factors analysis.

## 2.1. Aid Event Data Acquisition

The GDELT is a free and open news database. It has extracted and analyzed online news events published by world's media since its establishment in 1979 and is updated every 15 min. There are 58 fields in GDELT, which comprehensively record the actors, time of occurrence, the content of the event, place of occurrence, source of the event, and other information of the news event [32]. The GDELT fields involved in this paper are shown in Table 1. The Day field records the time of occurrence of the event. The data of events that occurred from 23 January 2020 to 31 October 2021 were selected for study in this paper. Then the events containing "COVID-19" in the SOURCEURL field were filtered as aid events. The Actor1CountryCode field and the Actor2CountryCode field record the country codes of the two actors of the event, separately. The aid events selected in this paper required that the two actors were not identical and not empty. The GDELT database defines an event as a single act generated by actor 1 to actor 2. Then an event in the type of actual aid could be expressed as an actual aid provided by actor 1 to actor 2. An event of the type of aid request could be expressed as a request for aid from actor 1 to actor 2. An event of the type of verbal aid could be expressed as a verbal promise of aid made by actor 1 to actor 2. Therefore, the aid in GDELT could be classified into three types: verbal aid, aid request, and actual aid. The EventRootCode and EventBaseCode fields record these types of events. An event with "07" in the EventRootCode field is an actual aid event. The events with the value "023" and "033" in the EventBaseCode field are aid requests and verbal aid events, respectively. On this basis, 21119 actual aid events, 489 aid request events and 1814 verbal aid events were obtained through filtering. International aid in this paper was referred to as actual aid. The three letter codes of actors involved in this paper (ISO alpha-3) and their corresponding countries are shown in Table A1 in the Appendix A. The percentage of the number of international aid events provided by each country is shown in Figure 2. To make the paper's conclusions more reliable and stable, the four countries that provide the highest number of actual aid events, China, the United States, the United Kingdom, and Canada, were selected for the study.

**Table 1.** The GDELT field names and meanings involved in this article.

| Id | Name | Meaning |
|----|------|---------|
| 1 | GlobalEventID | Unique identifier of the event |
| 2 | Day | The occurrence date of the event |
| 3 | Actor1CountryCode | CAMEO country code for actor 1 |
| 4 | Actor1Type1Code | Type of actor 1 |
| 5 | Actor2CountryCode | CAMEO country code for actor 2 |
| 6 | EventRootCode | Two-word code for event types |
| 7 | EventBaseCode | Three-word code for event types |
| 8 | SOURCEURL | Source URL of the event |

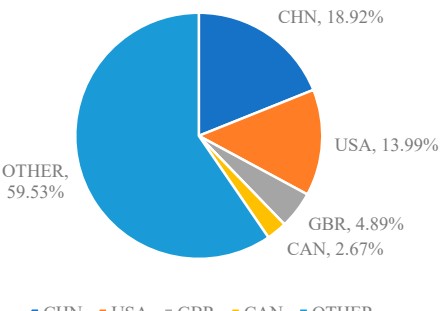

**Figure 2.** Proportion of events in which countries provided international aid.

*2.2. Methods*

2.2.1. Characteristics Analysis

The relevant characteristics of international aid during the COVID-19 pandemic were defined as follows:

1.  The characteristics of number of aid events. The number of the actual aid events provided by a donor country per day was considered as the number of aid events per day of the donor country;

2.  The characteristics of aid flow. The country in which actor 1 of each actual aid event in GDELT is located was considered as the donor country. The country in which actor 2 is located was considered as the recipient country. The aid allocation from the donor country to the recipient countries was considered as the aid flow from the donor country;

3.  The characteristics of aid source. The Actor1Type1Code field records the type of actor1, such as government (GOV), multinational companies (MNC), media (MED), business (BUS), education (EDU), non-governmental organizations (NGO), civilian (CVL), military (MIL), health (HLH), etc. The type of actor1 of the actual aid event can be considered as the source of aid provided by the country where Actor1 is located;

4.  The characteristics of response and cashing of aid events. An aid request event was regarded as one aid request from the recipient country to the donor country. Then actor 1 in the aid request events is the recipient country and actor 2 is the donor country. Multiple aid requests from the same recipient country to the same donor country on the same day were regarded as only one request. An actual aid from the donor to the recipient within 90 days of the request was considered a response to the request. A verbal aid event was considered as a verbal aid commitment from the donor country to the recipient country. Then in a verbal aid event, actor 1 is the donor and actor 2 is the recipient. If the donor provided actual aid to the recipient within 90 days of making the verbal aid, the verbal commitment was considered substantive and the actual aid event was considered a realization of that verbal aid.

In general, the shorter the response time of the donor to the recipient's requests for aid, the more responsive the donor is to the recipient's requests for aid. That is, the responsiveness of the donor to the recipient decayed with time. This is similar to Tobler's

first law of geography, which describes how the spatial action between two objects decreases with increasing distance, a phenomenon known as distance attenuation. Its basic forms such as exponential, power function and Gaussian function have been widely used, of which the power function type attenuation curve has been more commonly used [33]. Therefore, this paper proposed a time attenuation model based on a power function type attenuation curve to calculate the response degree and the cashing degree, which measured the responsiveness of donor countries to aid requests and the positivity of cashing in on the verbal aid, respectively. The calculation formula of response degree is as follows:

$$R_{ij} = \frac{P_{ij}}{Q_{ij}} \times C \left( \frac{\sum\limits_{k=1}^{P_{ij}} d_{ij}(k)}{P_{ij}} + 1 \right)^{-\alpha} \tag{1}$$

where $R_{ij}$ is the response degree of country $i$ to the aid requests of country $j$, $P_{ij}$ is the number of responses of aid requests of country $i$ to country $j$, $Q_{ij}$ is the total number of aid requests sent from country $j$ to country $i$, $d_{ij}(k)$ is the time interval between kth response to country $j$ of country $i$, $C$ is the decay constant, and $\alpha$ is the friction factor of distance decay. In this paper, we took $C$ as 1 and $\alpha$ as 1.5.

The cashing degree was calculated by the following formula:

$$F_{ij} = \frac{M_{ij}}{N_{ij}} \times C \left( \frac{\sum\limits_{k=1}^{M_{ij}} d_{ij}(k)}{M_{ij}} + 1 \right)^{-\alpha} \tag{2}$$

where $F_{ij}$ is the cashing degree by country $i$ to country $j$, $M_{ij}$ is the number of cashing from country $i$ to country $j$, $N_{ij}$ is the number of verbal aid from country $i$ to country $j$, and $d_{ij}(k)$ is the number of days between the kth cashing of verbal aid.

### 2.2.2. Regression Analysis of Influencing Factors

Quantitative analysis of influencing factors mostly adopts the mathematical statistics analysis method. That is, data mining and information extraction are performed on a large number of selected statistical data to explore the inherent regularity of data. Multiple regression models are empirical models that often use influencing factors as independent variables to explain changes in dependent variables, which assume a linear relationship between changes in dependent variables and independent variables in a specific region and over a specific period [34]. This paper used national panel data, and the general form of its regression model is as follows:

$$y_{it} = \beta_0 + \sum_{k=1}^{K} \beta_{kit} x_{kit} + \mu_{it} \tag{3}$$

where $i$ denotes an individual, $t$ is time, $y_{it}$ is the observed value of the $i$th individual in period $t$, $x_{kit}$ is the observed value of the $k$th independent variable for individual $i$ in period $t$, $\beta_{kit}$ is the parameter to be estimated, and $\mu_{it}$ is the random disturbance term.

Monthly aid data from January 2020 to October 2021 for the seven countries with the highest aid from China, the United States, the United Kingdom, and Canada were intercepted to study the impact of each influencing factor on aid. The 10 influencing factors as shown in Table 2 were selected for regression analysis, and the model in (3) was specified as follows:

$$\begin{aligned} aid_{ijt} = {} & \alpha_0 + \alpha_1 oth_{ijt} + \alpha_2 \ln pop_j + \alpha_3 \ln dis_{ij} + \alpha_4 hos_j + \alpha_5 con_{jt} + \alpha_6 die_{jt} \\ & + \alpha_7 \ln gdp_j + \alpha_8 \ln rgdp_j + \alpha_9 \ln tra_{ij} + \alpha_{10} vote_{ij} + \varepsilon_{jt} \end{aligned} \tag{4}$$

where $aid_{ijt}$ is the aid times of country $i$ to country $j$ in month $t$, $oth_{ijt}$ is the aid times of other countries except country $i$ to country $j$ in month $t$, $pop_j$ is the population of country $j$, $dis_{ij}$ is the geographic distance between country $i$ and country $j$, $hos_j$ is the number of hospital beds per 10,000 people in country $j$, $con_{jt}$ is the number of newly confirmed cases per million people in country $j$ in month $t$, $die_{jt}$ is the number of new deaths per million people in country $j$ in month $t$, $gdp_j$ is the gross domestic product (GDP) of country $j$, $rgdp_j$ is the real GDP per capita in country $j$, $tra_{ij}$ is the bilateral import and export volume between country $i$ and country $j$, $vote_{ij}$ is the UN voting consensus rate between country $i$ and country $j$, and $\alpha_0 \sim \alpha_{10}$ are the coefficients. To improve the model fit, population, geographic distance, GDP, real GDP per capita, and bilateral imports and exports were logarithmically processed based on previous experience. Moreover, all variables in this paper, except for pandemic data, were from the previous year or month due to the lagged effect of each influencing factor on aid [35]. If monthly data is not available, annual data was used instead. The current month data of new cases and new deaths in recipient countries were used because one month is long enough to ignore the lag effect of the pandemic on aid. The result of the calculation of the amount of verbal aid plus the amount of actual aid by the donor country was used as the dependent variable. Because both verbal aid and actual aid indicate the willingness of donors to allocate aid resources. To avoid the existence of multicollinearity between the variables affecting the regression results, a VIF test was conducted, and the results are shown in Table 3. The results proved that there was no multicollinearity between the variables.

**Table 2.** Meaning and source of independent variables.

| Independent Variable | Source | Interpretation |
| --- | --- | --- |
| Amount of aid from other countries | GDELT Database | Number of verbal and actual aids from countries other than the donor to the recipient country |
| Population | | The number of people in each country was logarithmic |
| GDP | World Bank Database | The gross national product of the recipient country was logarithmic |
| Real GDP per capita | | The Real GDP per capita of recipient countries was logarithmic |
| Geographical distance | The CEPII database | Distance between the capital or major city of the donor country and the recipient country, calculated by latitude and longitude, with logarithmic treatment |
| Number of hospital beds per 10,000 people | World Health Organization's official website | Number of hospital beds per 10,000 people in recipient countries |
| Number of new diagnoses per million population | COVID-19 database from the Center for Systems Science and Engineering at Johns Hopkins University [36] | Number of new diagnoses per 1 million people in recipient countries |
| Number of new deaths per million people | | Number of new deaths per 1 million people in recipient countries |
| Bilateral import and export volume | UN Commodity Trade Database | The total bilateral imports and exports of donor countries and recipient countries were logarithmic |
| Unanimous vote in the United Nations | United Nations Digital Library | The unanimous vote rate of the donor country and the recipient country in the UN General Assembly, if one party was absent or abstained, the vote would not be counted |

**Table 3.** Results of VIF test.

| Variable | VIF | 1/VIF |
|---|---|---|
| rgdp | 2.60 | 0.3840 |
| pop | 2.36 | 0.4232 |
| die | 2.15 | 0.4654 |
| con | 2.13 | 0.4698 |
| tra | 1.82 | 0.5504 |
| hos | 1.68 | 0.5961 |
| oth_aid | 1.53 | 0.6539 |
| gdp | 1.47 | 0.6817 |
| dis | 1.42 | 0.7054 |
| vote | 1.33 | 0.7540 |
| Mean VIF | 1.85 | |

## 3. Results and Analysis

### 3.1. Analysis of Aid Characteristics

3.1.1. The Analysis of the Characteristics of the Number of Aid Events

Since the time series of the number of global aid event numbers fluctuated frequently, this study used loess local smoother to analyze the stability of the time series of global daily aid event numbers. The fitted line is shown in Figure 3. The period from 23 January 2020 to 31 October 2021 was divided into four stages as shown by the dotted lines in Figure 3: the first stage (23 January 2020 to 7 April 2020); the second stage (8 April 2020 to 5 November 2020); the third stage (6 November 2020 to 17 February 2021); and the fourth stage (18 February 2021 to 31 October 2021). Based on the results of the stage division, it could be possible to have a clearer study of spatiotemporal changes in the relevant characteristics of these four countries.

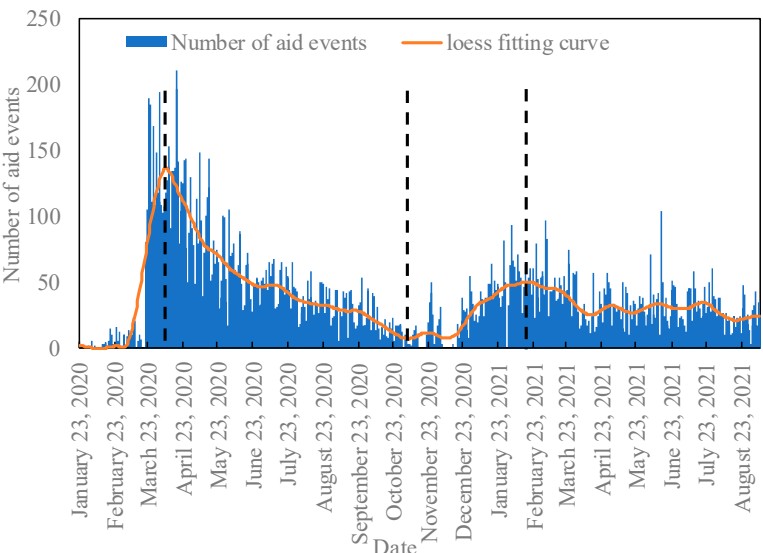

**Figure 3.** Time series and stages division results of the number of global daily aid events.

As seen in Figure 3, the number of aid events was first at a low level and then climbed sharply to a maximum in late February during the first stage. This is because the COVID-19 outbreak was first reported in China, while other countries have not yet experienced large-scale infection and spread of the virus, so there were few cases of aid. In late February 2020, Italy, South Korea, Iran, Singapore, and other countries broke out one after another. As more and more countries faced shortages of prevention supplies and medical equipment, inter-country aid activities began to surge. It reached a single-day maximum of 210 events on 16 April 2020.

The number of aid events in the second stage showed a gradual decline. As COVID-19 has spread globally, more and more countries and regions were experiencing outbreaks. In this situation, many companies, organizations, and countries have provided anti-pandemic materials such as masks and protective clothing or financial aid to the areas hardest hit by the pandemic. Therefore, the number of initial aid events in the second stage was at a high level. With the normalization of the pandemic, many countries have controlled the spread and development of the pandemic in their countries. At the same time, domestic efforts have been made to promote the production of materials such as masks and protective clothing, and a series of isolation policies have been adopted. Compared with the early stage of the pandemic, the lack of supplies and the medical experience was alleviated. As a result, global aid events began to gradually decrease. The amount of aid rebounded in the third stage, which may be related to the development of vaccines and increased vaccine aid. The number of aid events declined first in the fourth stage and then leveled off overall although there were occasional fluctuations.

### 3.1.2. The Analysis of Characteristics of Aid Flows and Sources

Figure 4 shows the characteristics of aid flows and the pandemic development in each country at each stage. It was intended not only to provide a more detailed description of the changes in the main aid recipients in China, the United States, Canada, and the United Kingdom at each stage, but also to analyze the relationship between these changes and the development of the pandemic in each country.

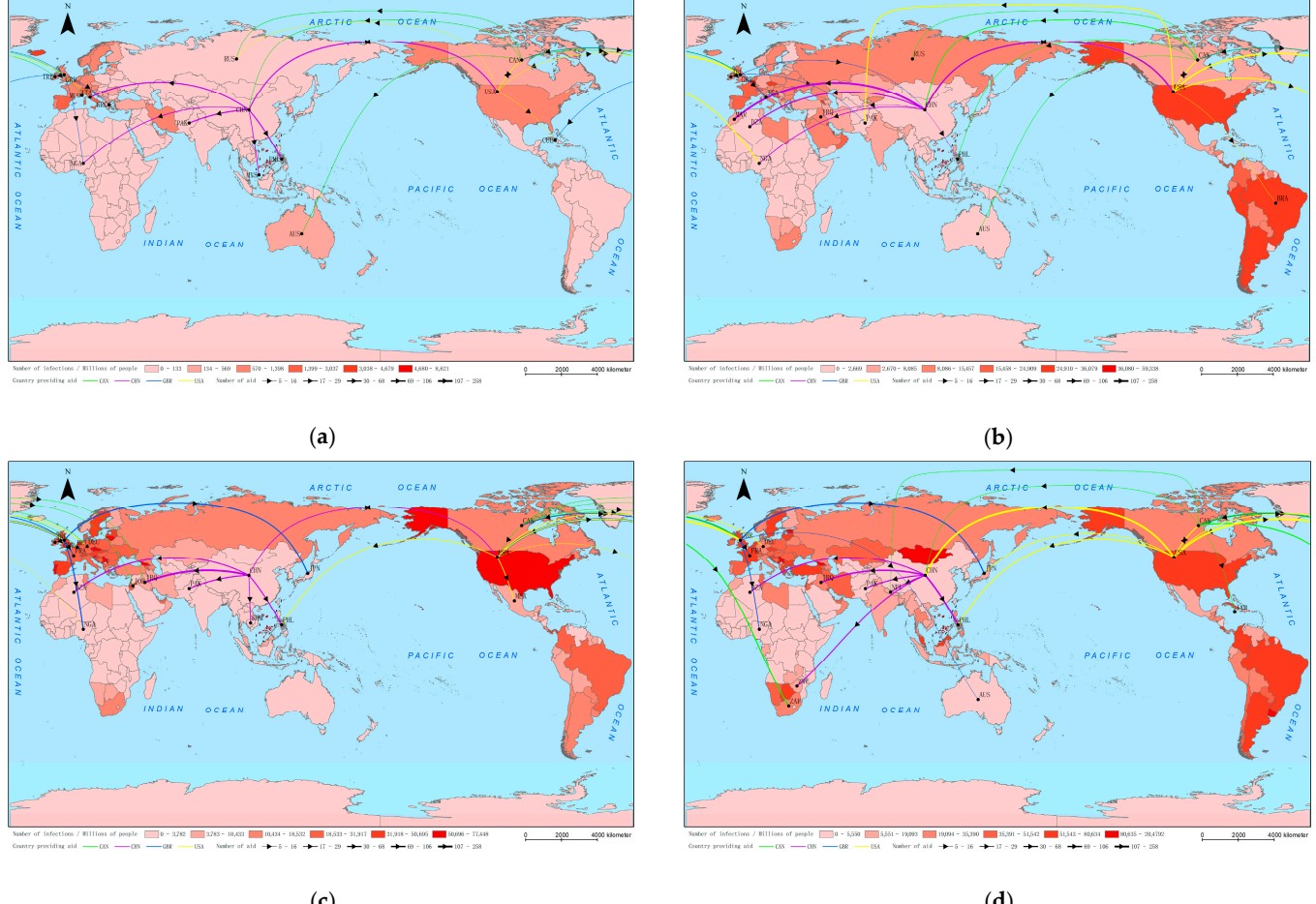

(**a**)  (**b**)

(**c**)  (**d**)

**Figure 4.** The main aid flows from the four countries and the number of new cases (per million) in each country of the world, divided by stage: (**a**) stage 1; (**b**) stage 2; (**c**) stage 3; and (**d**) stage 4.

As seen overall in Figure 4, aid from China, the United States, Canada, and the United Kingdom tended to flow to areas with severe outbreaks and neighboring countries. China and the United States have provided the largest aid at each stage. Of these, China has placed more emphasis on aid to countries in Asia and Africa. The United States and Canada have placed more emphasis on aid to countries in Asia, Europe, and North America. The United Kingdom has placed more emphasis on aid to European and North American countries. Asia has received focused aid from both China, the United States, and Canada. However, China and the United States have focused more on aid to South and Southeast Asian countries and Canada has focused more on East Asia. In addition, most of the countries China has aided were developing countries, while most of the countries that the United States, Canada, and the United Kingdom have aided in the high numbers belong to developed countries.

In the first stage, the pandemic has not yet broken out widely around the world, and the countries with serious outbreaks were mainly concentrated in Europe and North America. International aid during this stage was mainly concentrated in Europe, Asia, and North America. Among them, China has provided a great deal of aid, and the main recipient countries spread across Europe, Africa, Asia, and North America. The main recipient countries of the United States were concentrated in Europe and North America, while the United Kingdom and Canada provided less aid. In the second stage, as the outbreak took hold, the number of international aid events began to surge. At this stage, the main aid recipients of the United States were mostly distributed in West Africa, South Asia, and Europe. China mainly aided the United States and several countries in west and northwest Africa. The United Kingdom and Canada mainly provided aid to European countries and Asian countries. In the third stage, the global outbreak continued, but the amount of international aid began to decline. This is mainly because, after the accumulation of international aid and rich experience in fighting the pandemic in the second stage, each country had a certain ability to control the pandemic. China, the United States, and the United Kingdom provided the most aid during this stage. Of these, China has shifted its aid focus to Asia. The United States and Canada have provided aid primarily to European countries. The United Kingdom's main recipients were scattered across Europe, Asia, Africa, and North America. In the fourth stage, the number of new cases began to increase again in many countries. This may be due to the emergence of mutant viruses with greater transmission capacity, or lax government quarantine policies. At this stage, the donor countries were mainly China, the United States, and Canada. Among them, China mainly aided countries with serious outbreaks in Asia and Africa. The United States and the United Kingdom mainly aided Asian and European countries. Canada's major recipients were more dispersed and mainly aided the United Kingdom, the United States, and South Africa, where the outbreaks were more severe.

As shown in Figure 5, the sources of aid were relatively diverse and varied across countries. Aid from China was mainly provided by the government, accounting for 68.1%. Aid from the United States was dominated by government, multinational corporations, and media relief. Canada's aid sources were mainly the government and multinational corporations. However, 75.9% of the aid provided by the United Kingdom came from multinational companies and only 17.4% from the government. Governments might prefer to provide aid to countries with important economic and political ties, while multinational companies might provide aid to the relevant countries where the company operated. So different sources of aid would have different aid tendencies. This may be the reason why countries did not provide aid exclusively to countries with severe outbreaks.

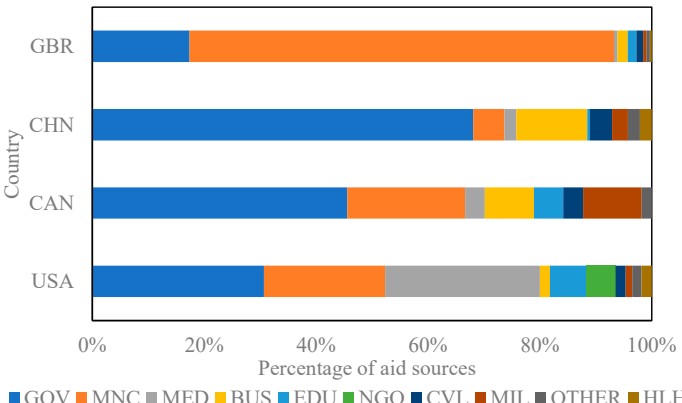

**Figure 5.** Percentage of sources of aid from the four countries.

### 3.1.3. The Analysis of Response Characteristics

Based on the results of the stage division, the response degree of China, the United States, the United Kingdom and Canada to aid requests from other countries was visualized, as shown in Figure 6.

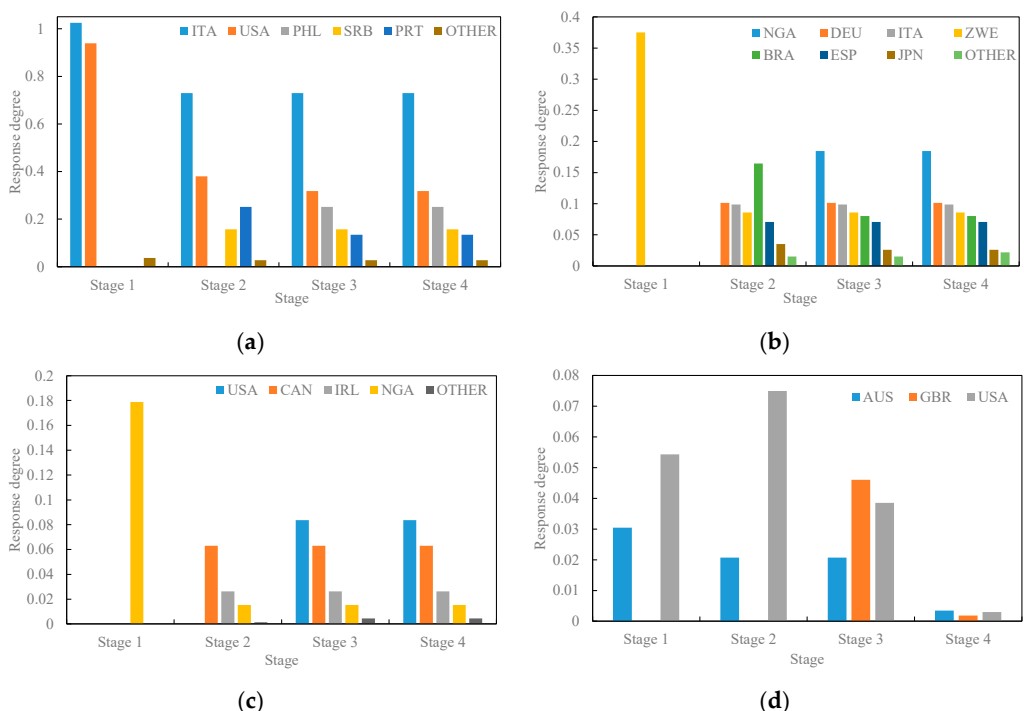

**Figure 6.** The response degree of each stage to aid requests from the four countries, divided by country: (**a**) CHN; (**b**) USA; (**c**) GBR; (**d**) CAN.

As can be seen in Figure 6, the majority of countries' aid requests were responded to from the second stage. The second stage was the moment when the COVID-19 pandemic became a major outbreak worldwide. This indicated that in the face of the crisis of COVID-19, most countries were hard hit economically and socially and urgently needed help from other countries.

It was also clear that the United States' aid requests were actively responded to positively by these three countries. In addition to this, China also gave more importance to Italy's aid requests, the United States to Brazil, Nigeria, and Zimbabwe, the United Kingdom to Nigeria, and Canada to Australia and the United Kingdom. China, the United States, the United Kingdom, and Canada had an average response degree of 0.242, 0.055,

0.033, and 0.029, respectively. This indicated that China has been more responsive to aid requests from other countries in comparison.

It can also be obtained from Figure 6 that the response degree to aid requests varied by donor countries at different stages. The response degree of most donors decreased from the second or third stage. For example, China's response to Ghana, etc., the United States to Zimbabwe, etc., and Canada to Australia, etc. decreased from the second stage. China's response to Portugal, etc., the United States to Brazil, etc., and Canada to the United States, etc. decreased from the third stage. The reason for this was that at first, aid from donor countries was in the form of protective materials such as masks, hand sanitizers, and medical equipment. With the successful production of the vaccine in several countries, vaccine aid has begun to be popular around the world. This increased the response time to aid requests [37].

### 3.1.4. The Analysis of Cashing Characteristics

Based on the results of the stage division, the cashing degree of the four countries to other countries was visualized. The results are shown in Figure 7.

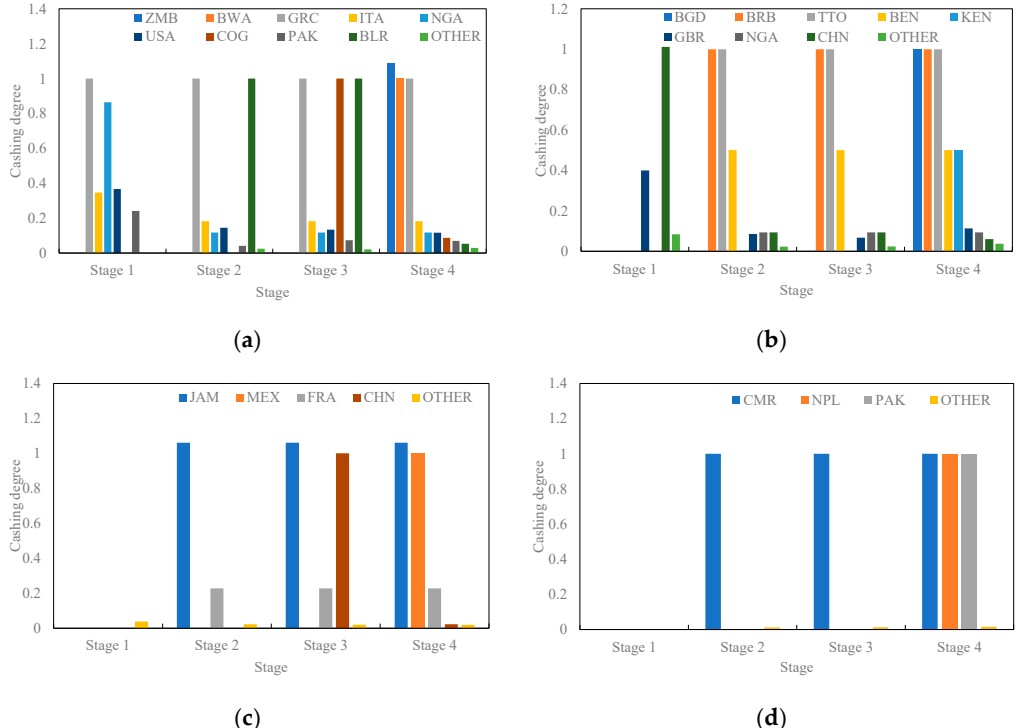

**Figure 7.** The cashing degree of each stage to aid requests from the four countries, divided by country: (**a**) CHN; (**b**) USA; (**c**) GBR; (**d**) CAN.

China, the United States, the United Kingdom, and Canada gave verbal aid to 43, 66, 31, and 13 countries, respectively. However, as can be seen in Figure 7, only a few countries honored their commitments. This showed that although China, the United States, the United Kingdom, and Canada gave a high amount of verbal aid, they did not honor all of their verbal commitments due to the donor countries' pandemic situation, economic development status, and political systems. The average cashing degree of China, the United States, the United Kingdom, and Canada were 0.20, 0.11, 0.14, and 0.24, respectively. China and Canada had a higher average cashing degree, while the United States and the United Kingdom had a lower one. This indicated that China and Canada were more motivated to cash in on their verbal aid commitments.

It can also be observed that China attached more importance to verbal aid commitments to Belarus, Congo, Greece, Nigeria, and Zambia. The United States attached more importance to Bangladesh, Barbados, China, and Trinidad and Tobago. The United King-

dom attached more importance to China, Jamaica, and Mexico, while Canada attached more importance to verbal aid commitments to Cameroon, Nepal, and Pakistan. From Figure 7, we could also see that several countries showed an upward or downward trend in the second and third stages of cashing in. During the first and second stages of the pandemic, China, the United States, the United Kingdom, and Canada also faced serious problems such as tight supplies for pandemic prevention and insufficient experience in fighting the pandemic, so had no time and materials to provide aid to other countries. Domestic outbreaks were effectively controlled after a series of measures were taken to strengthen the production of anti-pandemic materials and develop anti-pandemic policies. Only then did these countries step up their material aid to other countries, thus the cashing degree increased. The reason for the decrease in cashing degree was similar to the response degree and might be because vaccine aid increased the cashing time of verbal aid.

### 3.2. The Analysis of Influencing Factors to Aid Events Based on the Regression

Since there were both time-varying variables and fixed variables in the impact factors, the regression analysis model of the influencing factors was solved using the random effects model in the least squares method. Regression analysis of aid allocation characteristics and 10 relevant influencing factors for China, the United States, the United Kingdom, and Canada were conducted in 2020, and the results are shown in Table 4.

**Table 4.** Least squares regression results.

| Random-effects GLS regression | | | | Number of obs = 336 | | |
|---|---|---|---|---|---|---|
| Group variable: country | | | | Number of groups = 28 | | |
| R-sq: | | | | Obs per group: | | |
| | within = 0.2047 | | | | min = 12 | |
| | between = 0.5832 | | | | avg = 12.0000 | |
| | overall = 0.2762 | | | | max = 12 | |
| | | | | Wald chi2(10) = 147.8800 | | |
| | corr(u_i, X) = 0 (assumed) | | | Prob > chi2 = 0.0000 | | |
| aid | Coef. | Robust Std. Err. | z | P > \|z\| | [95% Conf. Interval] | |
| oth | 0.1031 | 0.0317 | 3.25 | 0.001 | 0.0409 | 0.1653 |
| lnpop | −4.5577 | 1.0581 | −4.31 | 0.000 | −6.6315 | −2.4840 |
| lndis | 3.7457 | 0.7926 | 4.73 | 0.000 | 2.1921 | 5.2992 |
| hos | 0.1220 | 0.0783 | 1.56 | 0.119 | −0.0315 | 0.2755 |
| con | 0.0001 | 0.0003 | 0.12 | 0.903 | −0.0006 | 0.0007 |
| die | 0.0058 | 0.0190 | 0.31 | 0.759 | −0.0314 | 0.0431 |
| lngdp | −0.8556 | 0.9778 | −0.88 | 0.759 | −2.7721 | 1.0609 |
| lnrgp | −5.6300 | 1.7782 | −3.17 | 0.002 | −9.1153 | −2.1447 |
| lntra | 3.7494 | 0.6876 | 5.45 | 0.000 | 2.4017 | 5.0971 |
| vote | 0.5315 | 3.0803 | 0.17 | 0.863 | −5.5057 | 6.5687 |
| _cons | 13.9306 | 30.3664 | 0.46 | 0.646 | −45.5864 | 73.4476 |
| sigma_u | | | 2.3647 | | | |
| sigma_e | | | 10.8841 | | | |
| rho | | | 0.0451 (fraction of variance due to u_i) | | | |

The aid refers to the number of actual aid plus the number of verbal aid provided by the donor country, the oth refers to the number of aid from other countries, the lnpop refers to the logarithm of the number of population in the recipient country, the lndis refers to the logarithm of the geographic distance between the major cities or capitals of the two countries, the hos refers to the number of hospital beds per 10,000 people in the recipient country, the con refers to the number of newly confirmed cases in recipient countries, the die refers to the cumulative number of confirmed cases in recipient countries, the lngdp and the lnrgdp refer to the logarithm of the recipient country's GDP and real GDP per capita respectively, the lntra and the vote refer to the logarithm of the bilateral trade volume and the unanimous rate of the UN General Assembly vote between the two countries, respectively.

From the regression results, the five variables of the number of aid events from other countries, population, geographical distance between the two countries, real GDP per capita, and GDP were statistically significant at the 95% confidence level. Since

Prob > chi2 = 0, all variables passed the joint test, meaning that overall, the independent variables exhibited a significant effect on the dependent variable. The adjusted $R^2$ was 0.27.

The results show that the number of aid events from other countries, the geographical distance between the two countries, and the bilateral trade volume, had a significant positive impact on the aid allocation of donor countries. This meant that holding all other variables constant, the further the two countries were from each other, the higher the bilateral trade, the more aid the recipient country received from the other countries, and the more aid the donor country provided to that recipient country. The reasons for this could be that there was competition in providing aid among donor countries, or that the countries receiving more aid had serious outbreaks themselves; and so, donors were likely to provide more aid to recipients who have received greater aid from other countries. In other words, there was a clear "bandwagon effect" [38]. The positive coefficient of bilateral trade volume indicated that donors also considered the factor of self-interest when making aid decisions to provide more aid to maintain and promote economic cooperation with trading partners. The data indicated that there was a tendency to provide aid to distant countries. First, this distance was constructed from the distance between the capitals or major cities of two countries. It was influenced by the actual size of the country and the geographical location of the major cities, rather than the real distance between the two countries. Second, the factor of geographical distance itself did not have much explanatory power. As stated in the previous findings, aid provided by donor countries was affected by the outbreak in the recipient country, although the donor would respond to the aid requests. Thus, the factor of geographic distance has limited influence on aid allocation driven by the pandemic [39]. From the perspective of the model itself, the dependent variable was a time series, while geographic distance was a fixed variable. Geographical distance therefore had little explanatory power for the time-varying aid allocation of aid donors. In summary, the use of geographic distance to measure trends in aid allocation during the pandemic required further validation.

The coefficient of population and real GDP per capita of the recipient country was negative. This indicated that the donor countries preferred to provide more aid to the recipient country with a smaller population and lower level of economic development. That was a clear "small country tendency". Aid from donors would tend to work better in smaller countries with smaller population and lower real GDP per capita. This was because countries with smaller population would receive more aid per capita if they provided the same amount of aid. At the same time, countries with lower real GDP per capita had less economic development, less well-developed health care systems, and lower level of medical care. Therefore, they had a more urgent need for aid. This suggested that donor countries also considered the factor of recipient country needs when making aid allocations.

Although aid events generated by the pandemic were studied, pandemic data such as the number of new cases and new deaths did not show a significant impact on aid in the data. That was because aid from the donor to the recipient country was not an ongoing event. As an example, the time series of Chinese aid to Morocco, the United States, Nigeria, and Italy in Figure 8 compared with the time series of new cases per month in each country, most of the aid events occurred at or near the outbreak of the pandemic in the recipient countries. This was because when the outbreak was just beginning, most countries lacked experience as well as adequate medical support equipment, personnel, and supplies. After a certain cushion of time after receiving aid supplies from the donor countries, the shortage of medical supplies in the recipient countries has been alleviated. The number of aid incidents stopped increasing due to the policies and attitudes of governments dealing with the pandemic. Although many countries were still experiencing an increase in new cases, they were no longer dependent on aid from donor countries. This fully indicated that the timing of the aid provided by the donor countries was related to the timing of the outbreak in each recipient country. In addition, it was one-sided and inaccurate to analyze the impact

of the pandemic on aid only through the new case time-series data and the new data of the aid event time series.

**Figure 8.** Number of Chinese aid events received and number of new cases per month in the four recipient countries, divided by country: (**a**) MAR; (**b**) USA; (**c**) NGA; (**d**) ITA.

## 4. Discussion

The findings of this paper proved that aid allocation by donors not only considered the needs of recipient countries but was also influenced by aid competition factors and national interests. This phenomenon may lead to irrational distribution of aid resources, which results in the final aid effect varying significantly from the set target. During the pandemic, an irrational aid allocation can lead to a more unbalanced distribution of medical resources, which may have little effect in advancing the global fight against the pandemic. This is a reminder that it is essential to allocate aid resources properly to make aid most effective. The following are some of our recommendations for aid policymakers:

1. Clarify the purpose of aid and the selection standards of recipient countries. First, the purpose of aid should be clearly defined and firmly established when making aid policies. Secondly, it is recommended to clarify the aid recipients before making relevant aid policies. Recipient countries can be classified or prioritized by taking into account their national conditions and the level of economic and political interaction between the two countries. In this way, the aid allocation is more likely to be free from the influence of other factors, while also making international aid more equitable and effective;

2. Balancing the needs of the recipients with the benefits of the donors. Self-interest and altruism are indispensable when aid is provided by donors. We propose to construct a system of indicators for aid allocation. This would allow donor countries to find a balance between meeting the donors' political and economic interests and humanitarianism when making aid allocations;

3. Enhancing the coordination role of multilateral institutions for aid allocation. Excessive concentration of aid or too much biased aid can lead to a situation where some countries receive too much aid, while some countries in urgent need of aid do not receive sufficient aid. There is an urgent need in this situation for a stronger role in macro aid coordination. It is essential to use multilateral institutions such as the

United Nations and the OECD to rationalize the layout and plan the distribution of aid size. This can avoid aid competition and herding effects that lead to excessive concentration of aid in certain recipients.

This study examined the spatial and temporal characteristics and influencing factors of aid in four aid donors from a geographic perspective. However, the aid data in this paper were only from the GDELT database. The media, on the other hand, generally report news content that is selective and biased. As a result, the GDELT database will contain far more news about large countries than about small countries, although the actual events in these small countries may be much more than that. To circumvent this problem, we selected the four countries that have provided the most aid as the objects of our study. In comparison, they had relatively adequate data. The relevant studies could be considered reliable from the perspective of these four countries only. However, although the database provides information about the time, place, and actors of aid events, it is relatively difficult to obtain the specific content of aid such as masks, protective clothing, and amounts. Therefore, it is not possible to examine the specific aid characteristics and content of each country. In the subsequent study, the specific content or amount of aid could be obtained by introducing the aid information published on the official websites of each country or other sources. The amount of aid is then used as the dependent variable to examine the influencing factors of aid allocation of donors, which may give us more insight into the aid bias of donors. It is also possible to study the characteristics of the tendency of aid provided by the governments and the changes in the content of aid with the development of the pandemic. This is because government aid is more reflective of the political links between the donor and recipient countries.

Second, the research in this paper was mainly conducted from the perspective of donor countries. Further research could be considered from the perspective of recipient countries, such as the characteristics of staged sources of aid received by each recipient country, the characteristics of responses to aid requests, and the cashing characteristics of verbal aid. Aid effectiveness and the impact of aid effectiveness on aid allocation could be also further studied from the perspective of recipient countries.

In addition, regarding the analysis of factors influencing aid allocation, several of the factors selected in this paper failed to obtain the expected results in the regression analysis. In particular, the distance between the major cities or capitals of the two countries as a measure of geographic distance between the two countries is affected by factors such as the size of the two countries. Regression of the time series of the number of new cases and deaths on the time series of the number of aid events also failed to produce a significant correlation. Subsequently, more reasonable geographical distance data and pandemic data can be introduced to quantitatively study the relationship between these factors and international aid.

## 5. Conclusions

Based on the GDELT database, this paper was conducted with China, the United States, the United Kingdom, and Canada as they have provided a higher amount of aid driven by the outbreak. First, the flow and source characteristics of aid from these four countries were analyzed. Then, the response to aid requests and the cashing characteristics to verbal aid of these four countries were analyzed. Finally, a regression analysis of multiple factors influencing the allocation of aid resources in these four countries was conducted. The conclusions are as follows:

1.  China, the United States, the United Kingdom, and Canada have tended to provide aid to countries with severe outbreaks and neighboring countries. China placed more emphasis on aid to Asian and African countries. The United States and Canada placed more emphasis on aid to Asian and European countries, and the United Kingdom placed more emphasis on aid to European and North American countries. However, the United States, the United Kingdom, and Canada favored aid to developed countries, while China favored aid to some developing countries. The sources of

aid in these four countries were different: China's aid was mainly provided by the government; the United States' aid was dominated by government, multinational corporations, and media relief; Canada's aid mainly came from the government and multinational corporations; the United Kingdom provided most of its aid from multinational corporations;

2.  The second stage of the COVID-19 broke out worldwide, and most countries were unable to fight the pandemic alone and asked for aid from large countries with advanced medical equipment and strong economic power. China was far more responsive to aid requests than the United States, the United Kingdom, and Canada. The United States' requests received positive responses from the other three countries. Canada and China were the most active in responding to verbal aid commitments, but these four countries did not fully cash in on all verbal aid due to their pandemic or other reasons;

3.  Although it was a humanitarian aid activity, the four donors would provide more aid to the recipient countries with higher bilateral trade volume due to economic interests. Moreover, aid from other countries to the recipient country would positively promote aid from the donors to recipient countries. Small countries with lower GDP per capita and smaller populations would also receive more aid from the donor country. That was, donor countries' aid had an obvious "bandwagon effect" and "small country tendency". Since aid from donor countries to recipient countries was mostly concentrated during the outbreak period, the time series of the pandemic did not have a significant impact on aid. However, the impact of the severity of the pandemic in recipient countries on the allocation of aid resources could not be ignored.

**Author Contributions:** Conceptualization: Y.Y., Y.Z. and J.L.; Data curation, Y.L. and X.L.; Formal analysis, Y.Y.; Funding acquisition, Y.Z. and J.L.; Investigation, Y.Y.; Methodology, Y.Y. and Y.Z.; Resources, Y.Z.; Software, Y.Y.; Validation, J.L., Y.L. and X.L.; Visualization, Y.Y. and Y.Z.; Writing—original draft, Y.Y., Y.Z. and J.L.; Writing—review & editing, Y.Y., Y.Z. and J.L. All authors have read and agreed to the published version of the manuscript.

**Funding:** This research was funded by the National Social Science Foundation of China under Major Project, solicited by the National Office of Philosophy and Social Science, under the title of Interdisciplinary Research on the Theory and Methodology of Geo-environmental Analysis in the Era of Big Data (No. 20&ZD138).

**Institutional Review Board Statement:** Not applicable.

**Informed Consent Statement:** Not applicable.

**Data Availability Statement:** The datasets generated during the current study are available in the zenodo repository, https://doi.org/10.5281/zenodo.6416526 (accessed on 6 April 2022).

**Conflicts of Interest:** The authors declare no conflict of interest.

## Appendix A

**Table A1.** The three letter codes (ISO alpha-3) of countries involved in this paper and their corresponding countries.

| Code | Country |
|------|---------|
| AUS | Australia |
| BEL | Belgium |
| BGD | Bangladesh |
| BLR | Belarus |
| BRA | Brazil |
| BRB | Barbados |
| BWA | Botswana |
| CAN | Canada |

**Table A1.** *Cont.*

| Code | Country |
| --- | --- |
| CHN | China |
| CMR | Cameroon |
| COG | Republic of the Congo |
| CUB | Cuba |
| DEU | Germany |
| DZA | Algeria |
| ESP | Spain |
| FRA | France |
| GBR | United Kingdom |
| GRC | Greece |
| IRL | Ireland |
| IRQ | Iraq |
| ITA | Italy |
| Code | Country |
| JAM | Jamaica |
| JOR | Jordan |
| JPN | Japan |
| KEN | Kenya |
| KHM | Cambodia |
| MAR | Morocco |
| MCO | Monaco |
| MEX | Mexico |
| MYS | Malaysia |
| NGA | Nigeria |
| NPL | Nepal |
| PAK | Pakistan |
| PHL | Philippines |
| PRT | Portugal |
| RUS | Russian Federation |
| SRB | Serbia |
| TTO | Trinidad and Tobago |
| USA | United States |

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
