# Peer review of "Analysis of Spatiotemporal Characteristics and Influencing Factors for the Aid Events of COVID-19 Based on GDELT"

_sustainability, doi:10.3390/su141912522_

Round 1

Reviewer 1 Report

Title: Analysis of spatiotemporal characteristicsand influencing factors for the aid events for COVID-19 based on GDELT

The authors have analysed the spatiotemporal characteristics and influencing factors for the aid events for COVID-19 using GDELT data source. The characteristics of spatiotemporal changes in the main aid flow, the responsive characteristics of the aid requests, and characteristics of verbal aids to cash are studied using spatial statistical analysis. The aid influencing factors were analysed using regression analysis. The authors proposed method of response degree and cashing degree based on time attenuation model.

The overall analysis looks interesting. The quick response of donor to the recipient request for the aid is desirable. The responsiveness of the donor to recipient decay with time. The authors proposed time attenuation model, which looks promising. However, the authors are suggested to compare their proposed time attenuation model with existing models and should prove that the proposed model is better than existing models.

I agree with authors that multiple regression models are empirical models that often use influencing factors as independent variables to explain changes in dependent variables and independent variables in specific region and over specific period of time. The authors rightly applied the regression to quantitatively analyse the influencing factors. The authors may explore and apply other data mining and machine learning techniques to analyse the influencing factors. 

Author Response

Thank you for your comments.The responses regarding your comments are attached. The revision marks in the paper have been marked in red, see Manuscript-revision for details. The final version of the revised paper can be viewed in Manuscript-accepted revision.

Reviewer 2 Report

The study proposes an analysis of the relationship between aid during the COVID-19 pandemic and a set of explanatory variables in several countries. Although the association between these variable may be of some interest, some important concerns emerged from reading of the paper.

Main comments
- Authors should better motivate how this study could be useful for policymakers and what policy implications it could have.

- The way in which the dependent variable is constructed is not entirely clear. If I understand correctly, if China onlny helped Zambia and India, there would be two observations in the dependent variable, i.e. aidchn-zam and aidchn-ind. If this is correct, this cannot be considered as a panel dataset, but instead it represents a network that should be treated properly and not through a random effects panel model. Otherwise, I did not understand well, so the definition of the dependent variable is not clear. 

- The results on the distance as regressor are definetly counterintuitive. The justification provided by the authors is not sufficient.

- Spatio-temporal models are an actual branch of regression models. In this kind of analysis, the authors do not take space into account (except for the inclusion of the distance in the set of regressors), but limit themselves to explain the temporal components through a panel model. Therefore, I would suggest to limit the reference to spatio-temporal data.

- It would be interesting to have as dependent variable the amount (in dollars) of aid from one country to another rather the number of aid events, which could be extremely limiting, especially since this has a low variation throughout the sample and biased estimates can be the natural consequence.

Minor comments

- Is there a literature for the definition of R in equation 1?
- The design matrix of regressors may suffer of multicollinearity, the authors should provide a correlation matrix between the variables to provide at least some evidence on this.

- In Table 3, I would suggest using fewer (and equal for each row in a column) decimals.

-Line 361, I think the authors mean "confidence level" and not "confidence interval".

Author Response

(The authors gave the same response as above.)

Round 2

Reviewer 2 Report

The authors have addressed all my comments.